# A Rare Case of Adverse Reaction to Metal Debris in a Ceramic-on-Ceramic Total Hip Replacement

**DOI:** 10.3390/jfb13030145

**Published:** 2022-09-08

**Authors:** Adriano Cannella, Tommaso Greco, Chiara Polichetti, Ivan De Martino, Antonio Mascio, Giulio Maccauro, Carlo Perisano

**Affiliations:** 1Orthopedics and Trauma Surgery Unit, Department of Ageing, Neurosciences, Head-Neck and Orthopedics Sciences, Fondazione Policlinico Universitario Agostino Gemelli IRCCS, 00168 Rome, Italy; 2Orthopedics and Trauma Surgery, Università Cattolica del Sacro Cuore, 00168 Rome, Italy

**Keywords:** total hip replacement, adverse reaction, metal debris, prosthesis, revision surgery, hip surgery, megaprosthesis

## Abstract

Adverse Reaction to Metal Debris (ARMD) is one of the most frequent complications after Total Hip Replacement (THR) and often a cause of surgical revision. This is true especially for implants with Metal-on-Metal (MoM) and Large Diameter Heads (LDHs), which are frequently used to improve stability and reduce the risk of dislocation. However, ARMD is not exclusive to MoM replacement, as it can also occur in other implants such Ceramic-on-Ceramic (CoC), Metal-on-Polyethylene (MoP), and Ceramic-on-Polyethylene (CoP). In these non-MoM implants, ARMD is not caused by the tribo-corrosion between bearing surfaces but, rather, by the fretting at the interface between neck and stem of dissimilar metals. A case of a severe ARMD that happened to a 73-year-old female patient with CoC bearing THR at the right hip is presented in this case report. In this case, the ARMD was misdiagnosed for over a year, resulting in the development of a massive pseudotumor. The treatment of choice was a two-stage revision with the implant of a hip megaprosthesis. After more than 2 years of follow-up, complete recovery of hip Range of Motion (ROM) and normalization of chromium and cobalt levels in blood and urine were achieved. Despite the relatively short follow-up period, this can be considered a successful treatment of a major and misdiagnosed ARMD in a non-MoM hip replacement.

## 1. Introduction

Total Hip Replacement (THR) is a common and successful orthopedic procedure. It has been estimated that in 2010 the prevalence of total hip replacement in the United States (US) population was 0.83%. Prevalence was higher in women and increased with age, reaching 5.26% in patients aged 80 and older. These estimates corresponded to 2.5 million individuals with total hip replacement in the US [1]. Over the years, to improve stability, survivorship, and clinical outcomes of the prosthetic implants, several solutions have been developed. Large Diameter Heads (LDHs) and dual-mobility implants were introduced to increase stability and reduce dislocation risk, while different bearing surface combinations (metal-on-metal, ceramic-on-ceramic, and highly crosslinked polyethylene) were introduced to decrease wear and wear-related complications and improve implant longevity [2,3,4]. Neck–stem modularity of the femoral component was introduced to allow independent control of not only length and offset, but also version. All these new solutions made it possible to obtain better surgical results and increase the number of protheses implanted yearly worldwide [5]. Along with the increase in prostheses implanted, the number of complications also rose. A relatively common complication is Adverse Reaction to Metal Debris (ARMD), a series of disorders due to the length of implantation and wear of the prosthetic components [6,7]. The incidence of ARMD has been estimated to be about 5% for Metal-on-Metal (MoM) THRs [8,9]. 

The rubbing of the prosthetic components on each other leads to the release of metal ions and small particles in the joint. This occurs due to tribo-corrosion at the articular surfaces or fretting corrosion at the modular junctions [10]. These products can cause local tissue irritation and exaggerated immune reaction leading to the formation of pseudotumors [10]. Pseudotumors are neoformations found on imaging and during surgery that develop around the prothesis because of fibrotic reaction, proliferation, and synovial fluid overproduction, and are usually classified according to the Imperial Classification by Hart et al. [11,12,13]. This classification, based on magnetic resonance imaging, consists of four grades (1, 2a, 2b, and 3), based on wall thickness, solid or fluid content, and lesion shape. 

The immunity response is called Adverse Local Tissue Reaction (ALTR) and has been classified into four categories according to the histological examination: macrophage-dominated, mixed macrophage–lymphocytic with or without hypersensitivity features, and granulomatous pattern [11]. These patterns are the product of the mitochondrial stress induced by metal ions phagocytized by macrophages, and the development of hypersensitivity in a pre-existing chronic inflammation setting driven by lymphocyte activation [12]. 

We presented the case of a 73-year-old woman with immune system pathology and with ARMD following previous THR with a ceramic-on-ceramic bearing and a Cobalt–Chrome (CoCr) modular neck coupled with a titanium alloy femoral stem.

## 2. Case Report

A 73-year-old woman came to our attention for right hip pain for over a year in the absence of trauma, associated with fever. She underwent bilateral THR, left side 7 years earlier while the right side 6 years earlier. The right hip prosthesis was implanted through an anterolateral approach. The components implanted were anatomic uncemented stem (Ti6Al4 V; SPS-Modular^®^, Symbios, Yverdon, Switzerland), modular neck (varus/long/retroverted; CoCr alloy), ceramic femoral head (Biolox Delta 36 mm/−4 mm, CeramTec GmbH, Plochingen, Germany), and press-fit 52 mm acetabular cup (Hilock, Symbios, Yverdon, Switzerland) with a neutral ceramic liner (CeramTec GmbH, Plochingen, Germany). The acetabular component was stabilized with two self-tapping screws (6.5 mm diameter × 20 mm length) and the proximal femur had a metallic cerclage. No pathological changes in the surgical scar were noticed. Swelling of the thigh with limited Range of Motion (ROM) of the right hip was found (flexion < 90°, extension < 20°, abduction < 40°, and adduction < 20°). The pain score on the Visual Analogue Scale (VAS) was 7 and the Oxford Hip Score (OHS) was 18. 

Other comorbidities included hypothyroidism, rheumatoid arthritis, and a Mixed Connective Tissue Disease (MCTD) with mostly signs of lupus for which she was followed by the rheumatology department of our hospital, however, she was not following the prescribed therapy for MCTD. 

Radiographs, CT scans, and an MRI of the right hip and femur were required. The hip X-rays showed the components in a proper position with no evidence of loosening and periprosthetic fractures (Figure 1). 

CT and the MRI confirmed the correct placement and osseointegration of the components. A cyst with lobular margins, thickened walls, and fluid density of 20 cm in the cranio-caudal length and 10 cm in the axial length was found. This lesion extended through the vastus medialis and vastus intermedius, dislocated the adductor muscles, infiltrated the gluteus minimus, and caused a diffused intrapelvic reaction dislocating the psoas (Figure 2 and Figure 3). This finding was a pseudotumor of 2B class according to the Imperial Classification, since the pseudotumor had thick walls, atypical fluid, hyperintense on both T1 and T2 images, and a lobulated shape [13]. 

The laboratory tests showed the following values: Hb 12.7 g/dL (12–16 g/dL); white cell count 7 × 10^3^ (4.5–8.5 × 10^3^); platelet count 250 × 10^3^ (150–400 × 10^3^) with C-Reactive Protein (CRP) of 38.0 mg/L (positive > 5 mg/L).

Preoperative serum and urinary Co and Cr levels were above the normal limit of 1.0 ug/L (serum) and 2.0 ug/L (urinary) (Table 1).

A two-stage revision surgery was planned due to the possible concomitant infection.

The first stage was performed through a posterolateral approach as per surgeon preference. A large pseudotumor surrounding the proximal femoral and the acetabulum was found. Due to its dimensions, the pseudotumor was aspirated with a needle and a total of 800 mL of gray-yellow stained fluid were drawn. Several samples of the pseudotumor were taken and sent for microbiological and histological examination (Figure 4).

After removal of the entire pseudotumor, the ceramic femoral head and the modular neck were removed; an extended trochanteric osteotomy was needed to remove the femoral stem. Upon removal of the acetabular component and its screws, additional gray-yellowish fluid was found. The femoral osteotomy was closed with two metal cerclages and a preformed antibiotic-loaded hip spacer was inserted (Figure 5).

Immediately after the surgery, the patient started an empiric antibiotic therapy with daptomycin 800 mg daily and meropenem 3 g daily (as she was allergic to cephalothin and cefoxitin). All the microbiological cultures and sonication fluid cultures were negative. The histological examination instead confirmed the diagnosis of ARMD, describing the presence of inflammatory tissue typical of a foreign body reaction with multinucleated cells, macrophages full of iron deposits, and histio-lymphocytic infiltrate. The patient was doing well during the 2 postoperative weeks, without surgical wound problems and not even neutrophilic leukocytosis. 

Therefore, the second stage was planned 17 days postoperatively with a proximal femoral replacement due to the bone loss of the proximal femur using the previous posterolateral approach. 

For the femoral reconstruction, a modular femoral body of 120 mm (three extension pieces of 30 mm, 40 mm, and 5 mm) with an intramedullary cementless stem extension of 13 × 120 mm (MUTARS^®^, Implantcast GmbH, Bextehude, Germany) was used. A cementless highly porous revision acetabular component (Delta Multihole TT, Lima Corporate, Villanova San Daniele, Italy), stabilized with four self-tapping screws (6.5 mm diameter × 20 mm length), was used for the acetabular reconstruction with a dual-mobility liner. A Trevira mesh (Implantcast GmbH, Bextehude, Germany) was wrapped around the prosthesis to reattach the soft tissues and reduce the dislocation risk. 

The patient was hospitalized for 6 days after surgery to monitor for early perioperative complications and any sign of infection. After discharge, the patient underwent clinical and radiographic outpatient follow-up at 2 weeks, 1, 3, 6, 12, and 24 months (Figure 6).

As shown in Table 1, already one month after the operation we had an important improvement in pain (from 7 preoperatively to 4 postoperatively), as well as for ROM and for hip function assessed with the OHS. Inflammatory markers along with serum and urinary Co and Cr levels underwent a progressive normalization, returning to the normal range.

The patient recovered painless full weight-bearing and returned to free activities of daily living within 3 months postoperatively and, after more than 2 years, she is still completely pain-free and conducts a normal daily life (Figure 7). Due to the presence of the same femoral implant on the left side, a strict follow-up with imaging and metal ion level is planned.

## 3. Discussion

ARMD has been reported mostly in MoM implants and especially in THR with Large Diameter Heads (LDHs). These implants are more susceptible to developing ARMD and as suggested by Reito et al., in 2016 in their metanalysis, the overall prevalence is 4.8%, a value that increases up to 21.3% when imaging and/or laboratory tests (with dosage of metal ions in blood and urine) are performed [1,2,3,4]. However, recent studies have shown that ARMD can be seen in non-MoM THR, as in our case [14,15,16,17,18,19,20,21,22]. In MoM LDH implants, the debris are produced mostly from the tribo-corrosion at the articular surface while, in non-MoM modular neck–stem implants of dissimilar metals, they are generated from fretting at the interface between neck and stem [5,6,7]. The initial modular neck systems, with titanium alloy neck and titanium alloy stem, did not offer sufficient tensile strength and clinical failures caused by implant fracture were reported; therefore, to increase implant rigidity and prevent fractures, cobalt chromium alloy necks were manufactured. Although this change led to a decrease in fractures of dual-taper modular components, retrieval analyses demonstrated corrosion and fretting related to micromotion at the mixed alloy neck/stem junction [23,24]. While there is still uncertainty on the gold standard for the preoperative management and the diagnosis of ARMD, some clarity has been obtained as to which are the intraoperative choices that can help reduce the risk of revision failure: the posterior approach, the revision of all components, the choice of >36 mm heads, and the use of MoP and CoP bearings [8].

This case adds to the existing literature and confirms that ARMD is a complication of THR to always be taken into consideration, especially in patients more prone to developing a rapid and strong immune response to metal ions and debris [12].

Our clinical case, as shown in Figure 8, confirms how ARMD, in implants with CoC bearing, develops from corrosion and repassivation at the neck–stem junction with release of chromium and cobalt particles and ions. The rarity of this occurrence is confirmed by the paucity of studies on ARMD in non-MOM implants in the literature, with only one additional case of revision of THR for ARMD with the same implant, in which the patient developed a pseudotumor of 8 × 9 × 10 cm and which has undergone a one-stage revision, obtaining a successful outcome like ours [25]. 

Clinical and functional outcomes (VAS, ROM, OHS) of our patient treated with a two-stage approach were comparable to the patient treated with a one-stage approach with the same femoral stem reported in the literature [9,26,27,28]. Therefore, it can be considered a successful treatment of a long-term misdiagnosed ARMD, which has reached such an advanced stage requiring a complex reconstruction. 

## 4. Conclusions

Considering the future growth of the number of implanted protheses, this report confirms that a thorough investigation of possible contraindication to any prosthetic implant should always be performed for each patient. Especially in those patients who, like the subject of our report, may have a pre-existing condition of an immune disorder that can positively influence the development of ARMD.

## Figures and Tables

**Figure 1 jfb-13-00145-f001:**
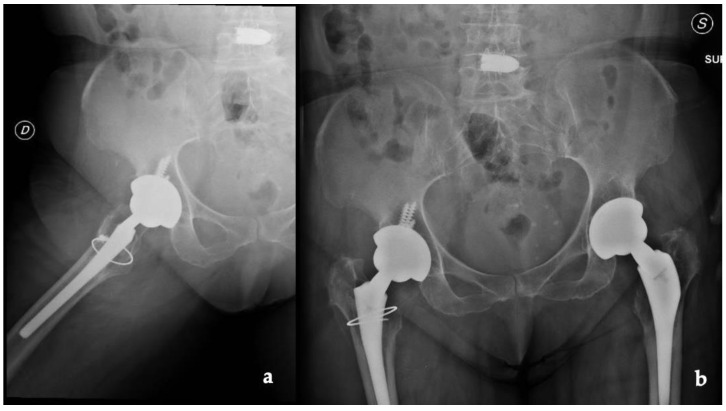
(**a**) Lateral and (**b**) anteroposterior view of the implant (D: right; S: left).

**Figure 2 jfb-13-00145-f002:**
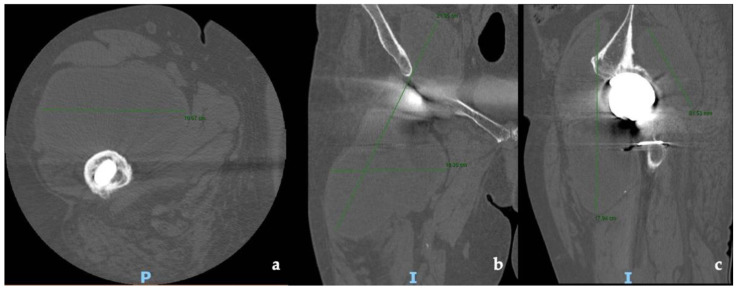
Preoperative CT scans of hip and thigh: (**a**) axial view; (**b**) coronal view; (**c**) sagittal view.

**Figure 3 jfb-13-00145-f003:**
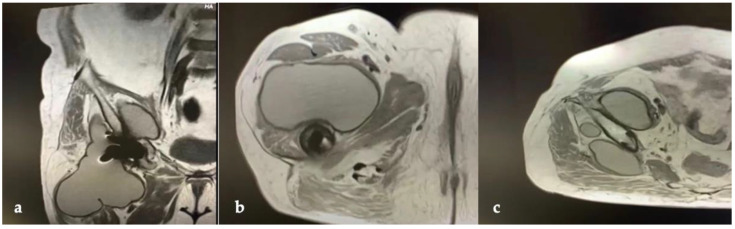
Preoperative MRI of hip and thigh: (**a**) coronal view; (**b**) axial view at femoral diaphyseal level; (**c**) axial view at acetabular level.

**Figure 4 jfb-13-00145-f004:**
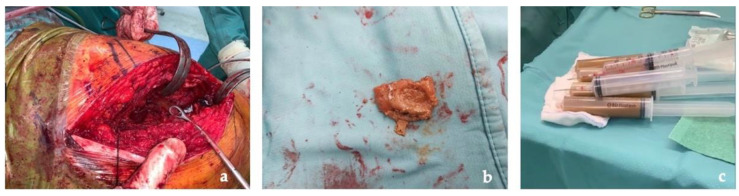
Intraoperative images: (**a**) pseudotumor identification at articular level; (**b**) pseudotumor capsule taken for histological exams; (**c**) fluid drained from the pseudotumor.

**Figure 5 jfb-13-00145-f005:**
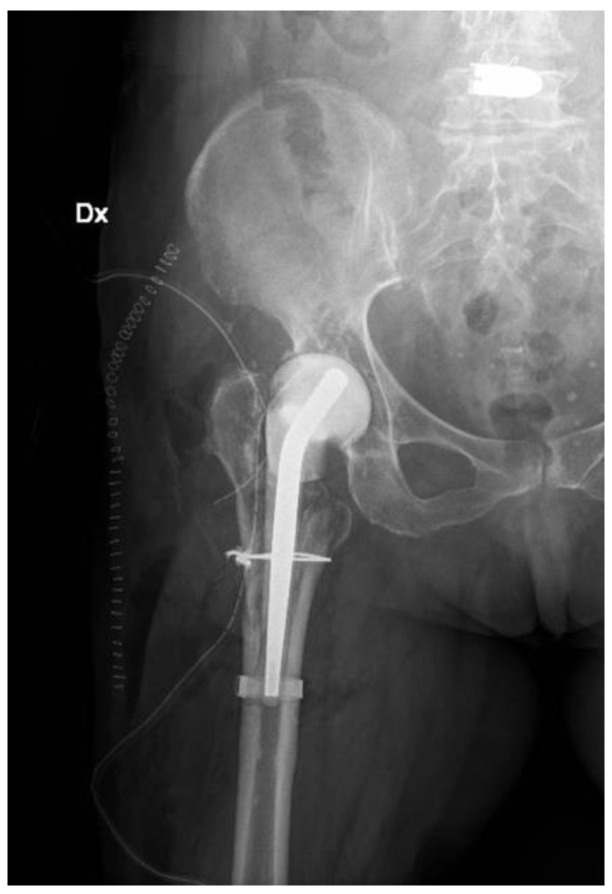
X-ray post first surgery with antibiotic spacer for hip (Dx: right).

**Figure 6 jfb-13-00145-f006:**
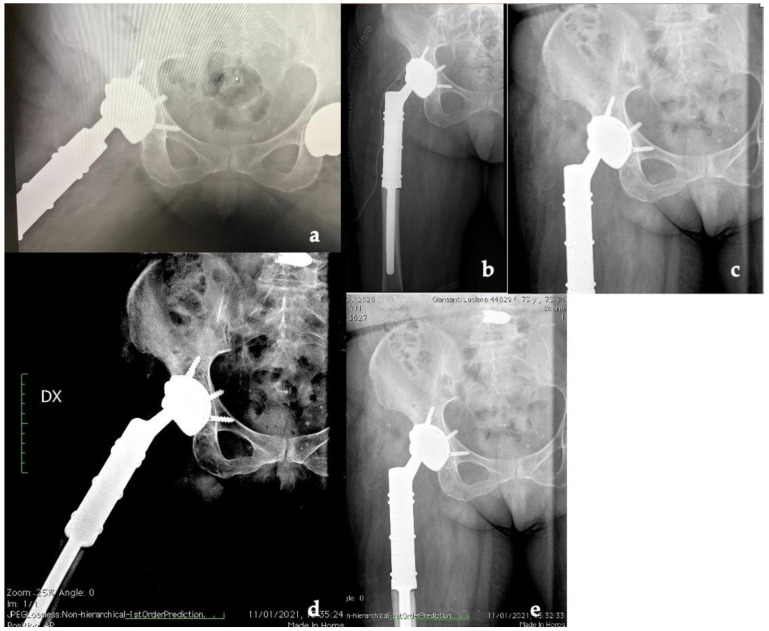
X-ray follow-up at (**a**) 1, (**b**) 3, (**c**) 6, (**d**) 12, and (**e**) 24 months (Dx: right).

**Figure 7 jfb-13-00145-f007:**
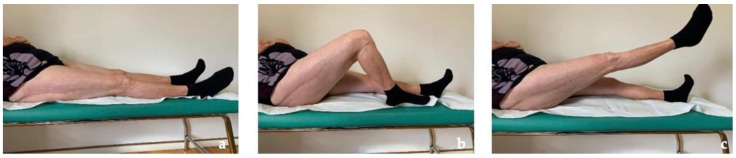
Clinical follow-up at 30 months with almost complete ROM of the right hip. (**a**) Neutral position. (**b**) Active knee flexion (**c**) Active hip flexion.

**Figure 8 jfb-13-00145-f008:**
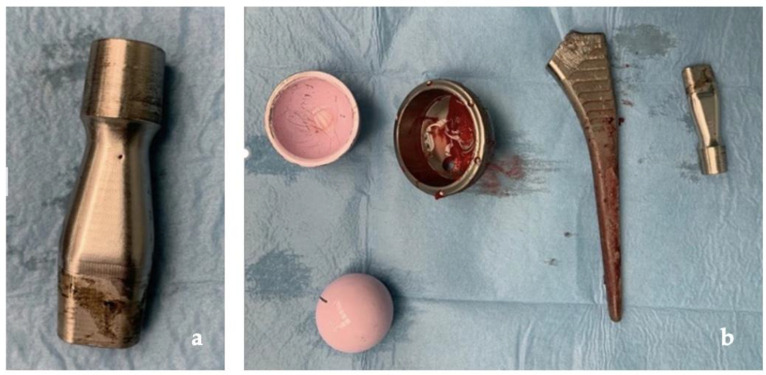
(**a**) Clear signs of tribocorrosion at the neck–stem junction; (**b**) prosthetic components removed.

**Table 1 jfb-13-00145-t001:** Clinical and laboratory follow-up. Visual Analogue Scale (VAS); Range of Motion (ROM); Oxford Hip Score (OHS); C-Reactive Protein (CRP); Cobalt (Co); Chrome (Cr).

		Pre-Op	1 Month	3 Months	6 Months	12 Months	24 Months
**VAS**	0–10 *	7	4	4	3	2	1
**ROM**							
** Flexion**	110–120° *	90°	95°	100°	105°	110°	110°
** Extension**	10–15° *	20°	20°	20°	25°	25°	25°
** Abduction**	45° *	40°	45°	45°	50°	50°	50°
** Adduction**	15–25° *	20°	20°	20°	25°	25°	25°
**OHS**	0–48 *	18	28	28	30	37	40
**CRP**	<5 mg/L ^†^	38 mg/L	<5 mg/L	<5 mg/L	<5 mg/L	<5 mg/L	<5 mg/L
**Cr blood**	<1.0 ug/L ^†^	13 ug/L	5.61 ug/L	2.98 ug/L	0.90 ug/L	0.78 ug/L	0.60 ug/L
**Co blood**	<1.0 ug/L ^†^	15 ug/L	6.73 ug/L	3.32 ug/L	1.47 ug/L	1.21 ug/L	0.90 ug/L
**Cr urine**	<2.0 ug/L ^†^	10 ug/L	6.31 ug/L	4.57 ug/L	2.88 ug/L	1.67 ug/L	1.17 ug/L
**Co urine**	<2.0 ug/L ^†^	12 ug/L	9.67 ug/L	7.38 ug/L	6.47 ug/L	4.31 ug/L	2.01 ug/L

* Range of value. ^†^ Normal value.

## Data Availability

The study data will be available upon request to the corresponding author (email: greco.tommaso@outlook.it).

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
