# Peer review of "A Rare Case of Adverse Reaction to Metal Debris in a Ceramic-on-Ceramic Total Hip Replacement"

_jfb, 2022, doi:10.3390/jfb13030145_

Round 1

Reviewer 1 Report

Specific points and suggestions for improvement of the manuscript are listed below.

General comments:
(1) The authors report a “Rare case” (title) and they further state (L182-5) “The rarity of this occurrence is ... only one additional case ... like ours”. Please see - Matharu et al. 2016 “Adverse reactions to metal debris occur with all types of hip replacement not just metal-on-metal hips: a retrospective observational study of 3340 revisions for adverse reactions to metal debris from the National Joint Registry for England, Wales, Northern Ireland and the Isle of Man”. BMC Musculoskelet Disord. 17(1):495. doi: 10.1186/s12891-016-1329-8. PMID: 27955657; PMCID: PMC5153685.

Maybe not all cases in this review are as described by the authors, but non-MoM ARMD were reported before. This could be added in the discussion.

Specific comments:

Abstract:

- L22 – please use full term “range of motion” (ROM), as it is used for the first time.

Case report:

- L103-8: most of the data already appear in table 1. I am not sure it needs to be repeated in the text. At least add here a reference to table 1.

Discussion:

- L166: “Reito et al.”, please add year.

- L180: “the figure 8” – “the” is redundant, please delete.

Tables:

- Table 1: please add normal range for parameters (appear in L103-8)

Figures:

- Figure 6: please label the images A, B, C etc. and refer accordingly in the legend.

Author Response

Specific points and suggestions for improvement of the manuscript are listed below.

Thanks to the reviewer for the careful review and suggestions.

General comments:
(1) The authors report a “Rare case” (title) and they further state (L182-5) “The rarity of this occurrence is ... only one additional case ... like ours”. Please see - Matharu et al. 2016 “Adverse reactions to metal debris occur with all types of hip replacement not just metal-on-metal hips: a retrospective observational study of 3340 revisions for adverse reactions to metal debris from the National Joint Registry for England, Wales, Northern Ireland and the Isle of Man”. BMC Musculoskelet Disord. 17(1):495. doi: 10.1186/s12891-016-1329-8. PMID: 27955657; PMCID: PMC5153685.

Maybe not all cases in this review are as described by the authors, but non-MoM ARMD were reported before. This could be added in the discussion.

Thank you for the very appropriate suggestion. The particularity and rarity of our case referred to the specific type of implant. But the suggested review is very appropriate and added to the discussion (Line 180).

Specific comments: 

Abstract:

- L22 – please use full term “range of motion” (ROM), as it is used for the first time.

Answer: thank you for suggestion. Modified as suggested (line 23). 

Case report:

- L103-8: most of the data already appear in table 1. I am not sure it needs to be repeated in the text. At least add here a reference to table 1.

Answer: thank you for suggestion. The text has been modified and the reference to Table 1 added.

Discussion:

- L166: “Reito et al.”, please add year.

Answer: thank you for the suggestion, the year has been included (line 177).

- L180: “the figure 8” – “the” is redundant, please delete.

Answer: thank you for the suggestion, the text has been modified (line 197).

Tables:

- Table 1: please add normal range for parameters (appear in L103-8)

Answer: thank you for the suggestion, normal ranges were included.

Figures:

- Figure 6: please label the images A, B, C etc. and refer accordingly in the legend

Answer: thank you for your suggestion, all figures have been revised as per your suggestion.

Reviewer 2 Report

The authors present an interesting case. Nevertheless, the article should be revised.

The introduction should be more specific about the stems with a modular neck module. How often have such cases been observed? Are there differences in modular systems with the same and different materials of the stem and neck? 

Why was a proximal femoral replacement performed during revision and not a modular stem? How was the gluteal musculature reinserted? Was insufficiency of the abductors observed?

What was the inclination of the acetabular component pre- and postoperatively? What was the stem version? Was there an increased stress on the modular stem in this case?

Why is the posterior approach to be favored in revision? Why does it lead to a supposedly lower revision rate? Doesn't it lead to an increased dislocation rate?

Why was the approach changed during revision?

The discussion should be more detailed. 

Author Response

The authors present an interesting case. Nevertheless, the article should be revised.

Thanks to the reviewer for the careful review and suggestions.

The introduction should be more specific about the stems with a modular neck module. How often have such cases been observed? Are there differences in modular systems with the same and different materials of the stem and neck? 

Answer: thanks to the reviewer for the suggestion. The discussion has been expanded, detailing the reasons that led to the introduction of modular neck systems, the evolution of these systems and the advantages/disadvantages of coupling different materials (Line 183-189).

Why was a proximal femoral replacement performed during revision and not a modular stem? How was the gluteal musculature reinserted? Was insufficiency of the abductors observed?

Answer: Thank you to the reviewer for the suggestion, which gives us the opportunity to better specify our surgical choice. Although the use of a modular revision stem was considered, the senior surgeon's choice, given his experience in the use of megaprosthesis, was proximal femoral replacement. During surgery an abductor deficit with extensive necrosis was observed, the remaining muscle was reinserted and anchored to the prosthesis using TREVIRA mesh (Line 152-154).

What was the inclination of the acetabular component pre- and postoperatively? What was the stem version? Was there an increased stress on the modular stem in this case?

Answer: Thank you to the reviewer for the comment and the request to clarify our case report. The acetabular inclination angle changed from 56° preoperatively to 51° postoperatively. The stem version could not be calculated as the patient did not have a total femur CT scan of. There was no increased stress in modular stem.

Why is the posterior approach to be favored in revision? Why does it lead to a supposedly lower revision rate? Doesn't it lead to an increased dislocation rate? Why was the approach changed during revision?

Answer: Thanks to the reviewer for the suggestion. The approach was chosen by the senior surgeon, based on his surgical experience balancing the risks (including the increased risk of dislocation) and benefits of the approach.

The discussion should be more detailed.

Answer: Thanks to the reviewer for the suggestion. The discussion was extended.

Reviewer 3 Report

In this case report, authors introduced a successful treatment of a long-term misdiagnosed ARMD. ARMD would happen not only in MoM implants but also some non MoM THR. This case report may be helpful for clinical practice. There are only some minor comment from me.

1) In Abstract, the full name of ROM should be given.

2) In this case, did the ARMD have relationship with the comorbidities (such as rheumatoid arthritis)? 

3) In the non MoM implants, will the wear debris of non-metallic interface occur similar symptom like ARMD? 

Author Response

In this case report, authors introduced a successful treatment of a long-term misdiagnosed ARMD. ARMD would happen not only in MoM implants but also some non MoM THR. This case report may be helpful for clinical practice. There are only some minor comment from me.

Thanks to the reviewer for the careful review and suggestions.

1) In Abstract, the full name of ROM should be given.

Answer: thank you for suggestion. Modified as suggested (line 23).

2) In this case, did the ARMD have relationship with the comorbidities (such as rheumatoid arthritis)? 

3) In the non MoM implants, will the wear debris of non-metallic interface occur similar symptom like ARMD?

Answer: thank you to the reviewer for these questions that allows us to further clarify the pathogenesis of the disease. As described by Ricciardi et al and Eltit et al (PMID: 26924942, PMID: 31417898), ARMD can have different patterns of development and different clinical symptoms in different types of implants. In MoM it is caused by a macrophage-dominated response triggered by the hypoxic stress mitochondrial response at the by cobalt ions, whereas in non-MoM it is probably mediated by the activation of a Th1 hypersensitivity reaction that is more likely in patients with a history of chronic inflammation or immune dysregulation typical of lupus and rheumatoid arthritis.

Round 2

Reviewer 2 Report

The authors revised their manuscript in accordance to the reviewer´s suggestions.